# Strategies to Target Chemoradiotherapy Resistance in Small Cell Lung Cancer

**DOI:** 10.3390/cancers16203438

**Published:** 2024-10-10

**Authors:** Tony Yu, Benjamin H. Lok

**Affiliations:** 1Department of Medical Biophysics, Temerty Faculty of Medicine, University of Toronto, 101 College Street, Toronto, ON M5G 1L7, Canada; 2Radiation Medicine Program, Princess Margaret Cancer Centre, 610 University Ave, Toronto, ON M5G 2M9, Canada; 3Department of Radiation Oncology, Temerty Faculty of Medicine, University of Toronto, 149 College Street, Toronto, ON M5T 1P5, Canada; 4Institute of Medical Science, Temerty Faculty of Medicine, University of Toronto, 6 Queen’s Park Crescent, Toronto, ON M5S 3H2, Canada

**Keywords:** cancer biology, cancer therapy, DNA damage, resistance, sensitization, small cell lung cancer

## Abstract

**Simple Summary:**

Small cell lung cancer (SCLC) is a deadly cancer that is treated by chemo- and radiotherapy, which work by damaging DNA. However, most SCLC patients will develop resistance to these treatments, resulting in a poor outlook with few effective treatment options available. This review summarizes our understanding of the causes of treatment resistance and the treatments which have been studied to overcome resistance. While there is still much to be understood, new methods being developed may improve our ability to study this disease and find effective treatments.

**Abstract:**

**Background:** Small cell lung cancer (SCLC) is a lethal form of lung cancer with few treatment options and a high rate of relapse. While SCLC is initially sensitive to first-line DNA-damaging chemo- and radiotherapy, relapse disease is almost universally therapy-resistant. As a result, there has been interest in understanding the mechanisms of therapeutic resistance in this disease. **Conclusions:** Progress has been made in elucidating these mechanisms, particularly as they relate to the DNA damage response and SCLC differentiation and transformation, leading to many clinical trials investigating new therapies and combinations. Yet there remain many gaps in our understanding, such as the effect of epigenetics or the tumor microenvironment on treatment response, and no single mechanism has been found to be ubiquitous, suggesting a significant heterogeneity in the mechanisms of acquired resistance. Nevertheless, the advancement of techniques in the laboratory and the clinic will improve our ability to study this disease, especially in patient populations, and identify methods to surmount therapeutic resistance.

## 1. Small Cell Lung Cancer

Small cell lung cancer (SCLC) is a lethal neuroendocrine carcinoma that accounts for about 15% of lung cancer cases [1]. Each year, there are approximately 250,000 new cases and 200,000 deaths globally, ranking it as the sixth most common cause of death from cancer [1]. Smoking is the main cause of SCLC development, with only about 2% of cases arising in never-smokers [2]. In addition, SCLC has a rapid doubling time, high growth fraction, and early metastasis, so the chance of early detection is low. This is especially the case because CT screening is not effective at detecting early-stage disease [3], and these patients are asymptomatic. Rather than the standard tumor, node, metastasis (TNM) staging system, SCLC is more commonly classified by the Veterans’ Administration Lung Study Group (VALSG) staging system into limited-stage and extensive-stage disease [4]. This staging system is based on whether the tumor can be safely confined to a single radiation port, and about two-thirds of patients will present with metastatic, extensive-stage disease.

### 1.1. Biology

#### 1.1.1. Recurrent Alterations

SCLC exhibits a high mutational burden and enrichment for CG to AT transversions, a mutational signature commonly observed in smokers [5,6,7,8]. In particular, the biallelic loss of the tumor suppressors tumor protein p53 (*TP53*) and RB transcriptional corepressor 1 (*RB1*) is nearly universal, and results in the dysregulation of the DNA damage response and cell cycle. This is thought to be the main driver of SCLC, and genetically engineered mouse models (GEMMs) with this dual knockout (KO) develop tumors resembling SCLC [9]. Other recurrent mutations have also been identified at lower frequencies through genomic and transcriptomic profiling of SCLC patient samples [5,6,7,8,10,11]. These include the loss of other tumor suppressors: tumor protein p73 (*TP73*), a *TP53* homolog; RB transcriptional corepressor like 1 (*RBL1*) and RB transcriptional corepressor like 2 (*RBL2*), which are paralogs of *RB1*; phosphatase and tensin homolog (*PTEN*); and Notch receptor (*NOTCH*) family genes. Loss of function mutations are also observed for genes involved in chromatin regulation such as lysine methyltransferase 2A (*KMT2A*), CREB binding protein (*CREBBP*), E1A binding protein p300 (*EP300*), and histone deacetylase 2 (*HDAC2*). In addition, activating mutations and amplifications of oncogenes have been observed in the MYC proto-oncogene (*MYC*) family genes (*MYC*, *MYCN*, *MYCL*), phosphatidylinositol-4,5-bisphosphate 3-kinase (PI3K) pathway genes, and RAS proto-oncogene (RAS) pathway genes. A small proportion of these mutations involved chimeric transcripts due to genomic rearrangements. SCLC is also clonally diverse, with multiple distinct clones contributing to intratumoral heterogeneity in each relapsed SCLC patient [11,12,13].

#### 1.1.2. Subtypes

Consistent with its neuroendocrine phenotype, most SCLC tumors highly express markers of neuroendocrine differentiation, such as insulinoma-associated protein 1 (INSM1), neural cell adhesion molecule 1 (NCAM1), chromogranin-A (CHGA), and synaptophysin (SYP), and are considered neuroendocrine-high [14]. However, some tumors express lower levels of these markers and are considered neuroendocrine-low. SCLC can be classified into four subtypes based on elevated expression of key transcription factors [15]: achaete-scute family bHLH transcription factor 1 (*ASCL1*) in SCLC-A, neuronal differentiation 1 (*NEUROD1*) in SCLC-N, Yes1 associated transcriptional regulator (*YAP1*) in SCLC-Y, and POU class 2 homeobox 3 (*POU2F3*) in SCLC-P. SCLC-A and SCLC-N have a neuroendocrine-high phenotype, while SCLC-Y and SCLC-P are neuroendocrine-low. However, studies have suggested that SCLC can exhibit significant plasticity and subtype switching, and patient tumors may represent a heterogeneous population of various subtypes as the tumor adapts and evolves [12,13,16]. GEMMs of SCLC have shed light on possible drivers of such events, though they remain to be fully validated in human disease. *MYC* amplification is thought to drive SCLC-A switching to SCLC-N and SCLC-Y, as GEMMs with *Trp53*/*Rb1* KO and hyperactive *MYC* develop invasive tumors of these other subtypes from primary SCLC-A tumors [17,18]. Furthermore, SCLC-A was found to correlate with *MYCL* amplification and negatively correlate with *MYC* amplification [19]. Similarly, activation of the NOTCH pathway can drive neuroendocrine-high to neuroendocrine-low transformation. GEMMs of SCLC overexpressing the intracellular domain of NOTCH1 develop neuroendocrine-low tumors from primary neuroendocrine-high tumors, and NOTCH pathway-mediated RE1 silencing transcription factor (*Rest*) and hes family bHLH transcription factor 1 (*Hes1*) expression represses neuroendocrine differentiation [20,21]. Indeed, SCLC-A negatively correlates with *NOTCH* family gene expression [8,20,22], and both SCLC-A and SCLC-N are enriched for delta like canonical Notch ligand 3 (*DLL3*), an inhibitory ligand of NOTCH proteins [14]. *YAP1* expression has also been found to drive neuroendocrine-high to neuroendocrine-low transformation. ASCL1 inversely correlates with YAP1 signaling [22], and in GEMMs of SCLC, expression of constitutively active *Yap1* resulted in tumor phenotype switching by upregulation of *Notch2* and downstream *Rest* and *Hes1* [21,23]. Moreover, *Yap1* is upregulated by NOTCH signaling, suggesting that these pathways feed into each other to promote neuroendocrine-high to neuroendocrine-low transformation [21]. As opposed to these, *POU2F3* expression is associated with chemosensory tuft cells [24], suggesting a different cell of origin, though whether SCLC-P can transform into other subtypes and vice versa is still unknown.

The existence of the SCLC-Y subtype is controversial. Though it was originally identified through transcriptomic profiling of SCLC cell lines and clinical samples, YAP1 protein expression in patient samples was found to be relatively low across tumors by immunohistochemistry [14]. A similar study classifying a biobank of circulating tumor cell-derived xenografts (CDXs) saw that SCLC-Y did not define a unique subtype of SCLC that was distinct from the others [25]. Given that SCLC-Y is more likely to retain *RB1*, studies have even suggested that cases of this subtype may belong to other cancers [26,27]. Another classification scheme replacing SCLC-Y with the inflammatory SCLC-I subtype has been proposed, characterized by low expression of the other transcription factors, an inflamed gene signature, infiltration by cluster of differentiation 8 (CD8)^+^/programmed cell death 1 ligand 1 (PD-L1)^+^ T-cells, and improved response to immunotherapy [28]. An alternative classification scheme based on transcriptomic analyses of clinical samples rather than transcription factors also identified immune-inflamed populations which respond best to immunotherapy, though these populations were neuroendocrine-high [29]. More work will be required to better classify SCLC and understand its phenotypic plasticity, as well as determine how this influences treatment outcomes or presents new therapeutic opportunities.

### 1.2. Treatment Options

In spite of advances in our understanding of its biology, SCLC is still treated clinically as a monolith, with differences based on disease stage [30,31,32]. Due to the lack of targetable driver mutations, the development of targeted therapies against SCLC has been slow. The high proliferation rate of SCLC sensitizes it to DNA damage, so the standard of care is a chemotherapy combination of a platinum-based DNA crosslinker, cisplatin or carboplatin, and the DNA topoisomerase II (TOP2) inhibitor, etoposide, across all stages. In rare cases of SCLC being detected early (T_1-2_N_0_M_0_), surgical resection followed by adjuvant chemotherapy is an option, though clinical evidence is limited [33]. For limited-stage disease, chemotherapy is given concurrently with thoracic radiotherapy. Patients who respond to this regimen are also given prophylactic cranial irradiation (PCI) to further improve clinical outcomes [34,35]. For extensive-stage disease, chemotherapy has remained the standard of care for over three decades. Addition of atezolizumab or durvalumab, which act as anti-PD-L1 agents, has received regulatory approval after demonstrating an overall survival (OS) benefit in large phase III trials [36,37]. Post hoc analyses further suggest that the addition of immunotherapy is effective against brain metastases [38,39]. While phase III trials of the anti-PD-1 antibodies, pembrolizumab and nivolumab, did not reach their primary endpoints of OS [40,41], a phase III trial of serplulimab showed similar efficacy to existing immunotherapies [42]. Interim results from the recent ADRIATIC phase III trial further demonstrated that consolidative durvalumab significantly improved OS and progression-free survival (PFS) in limited-stage patients, supporting its inclusion in the standard of care for these patients [43]. Additionally, a phase III trial by the Japanese Cooperative Oncology Group (JCOG) demonstrated a survival advantage when replacing etoposide with irinotecan [44], a TOP1 inhibitor. However, subsequent trials conducted by groups such as the Southwest Oncology Group (SWOG) were unable to replicate this improvement, though a different toxicity profile was observed [45,46]. The incorporation of radiotherapy for extensive-stage disease is also controversial, wherein a phase III trial investigating the addition of thoracic radiotherapy showed no statistically significant difference in OS at 1 year but a significant improvement at 2 years [47]. Similarly, while a European Organisation for Research and Treatment of Cancer (EORTC) study in the pre-MRI era found that addition of PCI for extensive-stage disease led to reductions in brain metastases with an increase in OS [48], no significant OS benefit was observed in a subsequent Japanese study mandating MRI screening [49].

While SCLC is quite sensitive to current first-line therapies, with objective response rates (ORR) of over 60% in patients with extensive-stage disease [50], more than 90% of patients will suffer from relapse within 2 years [51]. Generally, outcomes of relapsed disease differ based on the durability of response to first-line chemotherapy, with sensitive disease (commonly defined as duration ≥ 90 days) having better outcomes than resistant disease (duration < 90 days). Despite promising first-line outcomes, therapeutic options for the second-line and beyond remain limited and ineffective. Currently, the only FDA-approved second-line therapies for SCLC are topotecan, lurbinectedin, and tarlatamab. Topotecan, a TOP1 inhibitor, was approved following phase III trials demonstrating improved OS compared to best supportive care [52]. While topotecan was not superior to the combination of cyclophosphamide, doxorubicin, and vincristine (a chemotherapy regimen comparable to cisplatin and etoposide [53]), it did show improved symptom control [54]. Lurbinectedin, a DNA-alkylating agent, received accelerated approval after demonstrating efficacy against SCLC with a favorable toxicity profile in a phase II trial [55]. Tarlatamab, a bispecific T-cell engager targeting DLL3, similarly received accelerated approval due to favorable response rate and duration [56]. Confirmatory phase III trials are currently ongoing for both agents, pending full approval [57,58,59]. Patients with sensitive disease are typically retreated with platinum and etoposide, as a phase III trial demonstrated an improvement to the primary endpoint of PFS with carboplatin and etoposide compared to topotecan [60]. Similarly, a phase III trial of cisplatin, etoposide, and irinotecan demonstrated improved OS, PFS, and ORR compared to topotecan but had high rates of toxicity [61]. Several other agents like irinotecan, paclitaxel, temozolomide, and amrubicin have also shown activity against SCLC and are occasionally used in the second-line and beyond but are not superior to current therapies [62,63,64,65,66]. Immunotherapy with the anti-PD-1 antibodies nivolumab (alone or in combination with ipilimumab) and pembrolizumab have also demonstrated promising results in phase I/II trials and may be considered in the relapse setting [67,68], though a phase III trial of nivolumab alone as a second-line therapy did not reach its primary endpoint of OS [69].

## 2. Mechanisms of Resistance to DNA-Damaging Therapy in SCLC

One of the greatest challenges in SCLC treatment is the prevalence of treatment-resistant disease. The lack of targetable driver mutations limits the scope of effective targeted therapies, which is reflected in the current treatment paradigm consisting mostly of DNA-damaging chemotherapy. Although the rapid proliferation of SCLC causes increased sensitivity to DNA damage and replicative stress, recurrent disease no longer responds to this regimen, resulting in few effective therapeutic options available for treatment-resistant disease. This has spurred significant interest in understanding the mechanisms of therapeutic resistance in this disease, as well as identifying novel vulnerabilities and alternate approaches to treatment. Chemotherapy resistance has received more attention, given its broader application across SCLC treatment, though some work has also been conducted on radiotherapy resistance, and some mechanisms confer cross-resistance. The emphasis has been placed on enhancing DNA damage, such as by modulating repair and apoptosis pathways, as well as targeting SCLC differentiation and transformation pathways. Epigenetic regulation is another emerging area which has received increasing attention, and other mechanisms undergoing preclinical and clinical investigation include drug efflux, growth factor signaling pathways, metabolism, autophagy, and cell extrinsic factors such as the tumor microenvironment (TME). Some studies have also focused on sensitizers of chemo- and radiotherapy which are not linked to a particular resistance mechanism, such as cell cycle modulation. While new therapies have yet to reach the clinic, many new drugs are currently under investigation both preclinically and clinically (Table 1), including many combination therapies (Table 2), and this work has bolstered our understanding of SCLC biology.

### 2.1. Neuroendocrine-High to Neuroendocrine-Low Transformation

The pathways involved in the neuroendocrine-low differentiation of SCLC tumors (Figure 1), including YAP1, NOTCH, Wnt family (WNT), and *MYC*, have each been implicated in tumor progression and treatment resistance across various cancers. Indeed, neuroendocrine-low SCLC cells have been observed in GEMMs to be slower-growing but more chemoresistant [20,23]. As the master transcriptional regulator of the SCLC-Y subtype, YAP1 confers resistance to both chemotherapy and radiotherapy. For instance, YAP1 signaling has been correlated with drug efflux through breast cancer resistance protein (BCRP) and cancer cell stemness through CD133 [125,134]. It also feeds into the NOTCH pathway by upregulating *NOTCH2* expression [21,23]. Overexpression of YAP1 in the YAP1-low cell line H69 resulted in resistance to cisplatin, etoposide, and radiation [114,134]. On the other hand, overexpression of a dominant-negative form of YAP1 (lacking its transactivation domain) in the YAP1-high cell line H446 had the opposite effect of sensitizing cells to treatment. In these cells, treatment with a YAP1 inhibitor, verteporfin, also sensitized cells to chemotherapy (Table 2). Similarly, ex vivo culture of mouse SCLC cells with increasing doses of etoposide generated resistant cells with upregulated YAP1 [23], and ex vivo cultures of two CDXs with high YAP1 from a biobank of 39 CDXs show resistance to cisplatin [25].

Another pathway found to drive neuroendocrine-low differentiation and chemoresistance is the NOTCH pathway. In GEMMs, NOTCH signaling upregulated *Rest* and *Hes1* expression, two transcriptional repressors of neuroendocrine genes. Although tumor cells harboring *Rest* and *Hes1* upregulation were slower-growing, they were more resistant to treatment with cisplatin and etoposide and acted as part of the TME in providing trophic support to other tumor cells [20,21]. Indeed, recent studies in patient samples and preclinical models have found that neuroendocrine-low cells have the capacity to remodel the extracellular matrix (ECM) [135] and secrete extracellular vesicles containing ECM factors which support the growth of neuroendocrine-high cells [136]. Furthermore, such remodeling mediates vasculogenic mimicry, which increased the ability of cisplatin to reach the tumor, but conversely promoted resistance to cisplatin [135,137,138]. NOTCH1 and HES1 levels were also observed to be higher in the multidrug-resistant H69AR cell line, where siRNA knockdown (KD) of *HES1* sensitized cells to cisplatin and etoposide treatment [115]. Blocking NOTCH signaling with the NOTCH2/3 inhibitor tarextumab was able to delay onset of chemoresistance and synergize with chemotherapy in GEMMs [20], and inhibition of HES1 with FLI-06 was similarly able to synergize with chemotherapy in vivo [115]. In addition, NOTCH signaling has been shown to upregulate soluble guanylyl cyclase (sGC), a nitrous oxide (NO) sensor implicated in chemoresistance [139]. H196 cells with KO of a subunit of sGC are sensitized to etoposide treatment in vitro, and cell line-derived xenografts with this KO are more sensitive to cisplatin and etoposide. A randomized, multicenter phase Ib/II trial was undertaken comparing chemotherapy with or without tarextumab in 145 untreated patients with extensive-stage disease, but no significant improvement in PFS (5.5 vs. 5.5 months), OS (9.3 vs. 10.3 months), or ORR (68.6% vs. 70.8%) was observed and no biomarkers predicting efficacy were identified, resulting in the discontinuation of this drug’s development [70].

While the WNT pathway has not been directly implicated in neuroendocrine-low differentiation, ASCL1 has been shown to be antagonistic with WNT signaling in addition to YAP1 and NOTCH signaling [22]. Furthermore, whole-exome sequencing of treatment-naïve and relapse SCLC patient samples found that 80% of relapse samples had nonsynonymous mutations in canonical WNT pathway components that were not seen in naïve samples, resulting in higher levels of WNT activity [140]. Another study similarly identified mutations in APC regulator of WNT signaling pathway (APC), a negative regulator of canonical WNT signaling, across all relapse SCLC patient samples sequenced, as well as other WNT pathway mutations [11]. Correspondingly, cell lines with KD or KO of *APC* and elevated WNT activity were resistant to both cisplatin and etoposide and could be sensitized by re-expression of *APC* [140]. In addition, WNT signaling is known to upregulate the twist family bHLH transcription factor 1 (TWIST1) [141]. In patient-derived xenografts (PDXs) that were treated with cycles of chemotherapy to induce resistance, *TWIST1* was one of the most highly upregulated genes in resistant models [142]. However, while TWIST1 has been observed to mediate chemoresistance across several cancer types [143,144,145], the authors did not see chemosensitization with *TWIST1* KD. In a drug screen of SCLC cell lines and patient-derived cells, one of the top hits against cisplatin-resistant cells was the bromodomain and extraterminal domain (BET) family inhibitor, JQ1 [138]. JQ1 was predicted to target YAP1 signaling and the NOTCH, WNT, and transforming growth factor beta (TGFβ) pathways, and may be effective against tumors which leverage these resistance mechanisms. This agent may also be effective against cancer-associated fibroblasts (CAFs) which promote cisplatin resistance through TGFβ signaling [138], though this remains controversial as others have correlated CAFs with radiosensitivity and improved patient outcomes [146]. A combination between BET and mechanistic target of rapamycin kinase (mTOR) inhibition has also been proposed, as BET inhibitors can inadvertently promote tumor survival by upregulating the mTOR pathway [116].

MYC and MYCN, too, have been implicated in SCLC subtype transformation, and MYC is itself a target of both the NOTCH and WNT pathways [17,18]. Multiple studies have shown that high levels of MYC family proteins promote chemoresistance in SCLC both in vitro and in vivo [17,115,117,147,148,149], though MYC—and other classic oncogenes like RAS and Raf proto-oncogene (RAF)—were not found to confer radioresistance in SCLC [150]. In turn, MYC family proteins can promote NOTCH signaling to drive further tumor transformation [18]. Indeed, MYCN is capable of promoting chemoresistance through *HES1* expression, while NOTCH inhibition abrogates this effect [115]. MYC-high SCLC tumors have been shown to be sensitive to aurora kinase (AURK), ubiquitin specific peptidase 7 (USP7), mTOR, and dual PI3K/HDAC inhibition using in vivo models [17,147,148,151]. The AURKA inhibitor alisertib has demonstrated some clinical efficacy against SCLC both alone and in combination with paclitaxel (Table 1), and *MYC* expression was found to be a biomarker of treatment efficacy [71,72]. A double-blind phase II trial of paclitaxel with or without alisertib in 178 relapsed/refractory patients saw a marginal increase in PFS (3.32 vs. 2.17 months), with a greater benefit seen among 33 patients positive for MYC expression (4.64 vs. 2.27 months) [72]. A phase II trial has also recently been launched to test alisertib in extensive-stage disease progressing after chemoimmunotherapy (NCT06095505). On the other hand, a phase II trial of the AURKB inhibitor barasertib-HQPA in 15 relapsed/refractory patients did not show any efficacy, with an ORR of 0%. [73]. One recent study [152] looked at targeted MYCN inhibition using a peptide nucleic acid based on the *MYCN* antigene. The authors showed that this agent downregulated MYCN targets, including the mTOR pathway, and was effective as a monotherapy against a mouse xenograft of the multidrug-resistant cell line H69AR.

### 2.2. Cancer Stem Cells

Cancer stem cells are a very small subpopulation of a tumor which possess self-renewal capabilities and are important in propagating the tumor [153]. They are more treatment-resistant and promote tumor repopulation following therapy. As noted previously, YAP1 promotes the expression of one such stem cell marker, CD133, which correlated with radioresistance in SCLC (Figure 1) [134]. CD133 has also been shown to mediate resistance to etoposide, and CD133^+^ cells have been observed to accumulate in a more chemoresistant, post-treatment patient sample [154,155]. CD133 correlated with levels of AKT serine/threonine kinase (AKT) and BCL2 apoptosis regulator (BCL2), and increased tumor propagation potential, corresponding with its role in cell stemness [154]. Sarvi et al. noted that CD133^+^ cells had increased expression of neuropeptide receptors, and neuropeptide antagonists could preferentially target this population [154]. Another stem cell marker, CD87, has also been shown to mediate chemoresistance in SCLC. CD87 conferred resistance to both cisplatin and etoposide and correlated with levels of the drug efflux pump, multidrug resistance protein 1 (MDR1), and another stem cell marker, CD44 [155,156].

One of the drivers of cell stemness is the transcription factor SRY-box transcription factor 2 (SOX2). Through genomic and transcriptomic profiling, *SOX2* amplifications were identified in about 27% of patient samples and cell lines [6], as well as a proportion of relapsed patient samples [11]. Indeed, the key transcriptional regulator of the SCLC-A subtype, ASCL1, upregulates expression of *SOX2*, and this is important in both tumor establishment and MYC-dependent subtype regulation [19,157]. Upregulated *SOX2* expression is correlated with both cisplatin and etoposide resistance in SCLC cell lines, and the signal transducer and activator of transcription 3 (STAT3) inhibitor napabucasin was shown to downregulate SOX2 and MYC, thereby resensitizing cisplatin-resistant cells both in vitro and in vivo (Table 2) [117,158,159]. Strikingly, an extensive-stage patient with a *SOX2* amplification responded well to the Hedgehog inhibitor sonidegib in combination with chemotherapy in a phase I trial, remaining progression-free after 27 months on maintenance sonidegib despite the trial having a median PFS of only 5.5 months (Table 1) [74]. Hedgehog is known to regulate cell stemness through SOX2, suggesting that this signaling pathway may be involved in SCLC chemoresistance. However, a randomized phase II trial of chemotherapy with or without the Hedgehog inhibitor vismodegib in 155 untreated patients did not demonstrate a significant improvement in PFS (4.4 vs. 4.7 months) or OS (9.8 vs. 9.1 months), though selection for *SOX2* expression may improve outcomes [75].

### 2.3. Growth Factor Signaling

The PI3K/AKT/mTOR pathway is a critical signaling pathway for cell growth and proliferation which has been found to be dysregulated in nearly all cancers, including SCLC (Figure 1) [5,6,7,8,10,160]. Indeed, it is upregulated downstream of several key pathways in SCLC, including WNT and MYC signaling [151,152], and has been found to form a positive feedback loop with SOX2, implicating this pathway in cell stemness [158,159]. Genomic and transcriptomic analyses found that PI3K/AKT was upregulated in chemoresistant SCLC cell lines and patient samples [161] and that mTOR signaling was upregulated in a panel of chemoresistant PDXs [149]. Interaction of PI3K/AKT with SOX2 also promoted resistance to both cisplatin and etoposide [158,159], and inhibitors of PI3K and mTOR reversed this chemoresistance and synergized with chemotherapy [161,162]. Furthermore, upregulation of the PI3K pathway and tyrosine kinase signaling have been observed to mediate resistance to chemotherapy and radiotherapy by promoting adhesion of SCLC cells to the ECM through integrins [162,163,164,165]. Correspondingly, inhibitors of PI3K, mTOR, and epidermal growth factor receptor (EGFR) could reverse therapeutic resistance. Inhibitors of the PI3K pathway also abrogated radioresistance by promoting glucose-6-phosphate dehydrogenase (G6PD) degradation and upregulating the production of reactive oxygen species, which synergized with radiation both in vitro and in vivo [118]. In addition, a 2022 study [119] found that an inhibitor of exportin 1 (XPO1), selinexor, synergized with chemotherapy to inhibit tumor growth in PDXs of chemoresistant relapse patients. Through transcriptomic analyses, the authors identified the PI3K/AKT/mTOR pathway as being upregulated by XPO1 and confirmed that selinexor inhibits the activity of this pathway. While several clinical trials of PI3K (NCT02194049) and mTOR [73,76,77,78] inhibitors have not demonstrated significant activity (Table 1), patient selection for PI3K pathway upregulation and combination with other chemotherapeutics may improve the efficacy of targeting this pathway. A phase II trial was also initiated for selinexor in relapsed SCLC, but the trial was terminated due to slow patient accrual (NCT02351505).

Downstream of PI3K signaling, hypoxia inducible factor 1 (HIF-1) has been found to be upregulated [161]. The HIF-1 pathway is a key survival mechanism activated by hypoxic conditions and is involved in resistance to both chemotherapy and radiotherapy across many cancers. While downregulation of HIF-1α via KD of the long non-coding RNA HIF1A-AS2 sensitizes H69AR cells to doxorubicin [166], the role of the HIF-1 pathway and hypoxia in resistance to radiotherapy and other chemotherapeutic agents in SCLC remains to be elucidated. On the other hand, PTEN is a negative regulator of the PI3K pathway which acts in direct opposition to PI3K. Downregulation of PTEN activates the PI3K pathway and promotes chemoresistance in SCLC [167], and mutations in PTEN have been shown to mediate differences in radiosensitivity [168].

### 2.4. DNA Damage Response

Given that current SCLC therapies focus on DNA damage, cells can acquire resistance by bolstering their DNA damage response to repair damage and avoid lethality (Figure 2). One of the key pathways implicated in SCLC is the ATR serine/threonine kinase (ATR)/checkpoint kinase 1 (CHK1) pathway, which is critical for activating both the S and G_2_/M checkpoints and promoting DNA repair. Transcriptomic and proteomic profiling have shown that levels of both ATR and CHK1 are higher in SCLC than in non-small cell lung cancer (NSCLC) [120,169,170]. Preclinical and clinical data indicate that inhibitors of ATR are effective against SCLC tumors both alone and in combination with second-line therapies [79,169,171,172]. A proof-of-concept phase II trial of the ATR inhibitor berzosertib in combination with topotecan in 25 relapsed SCLC patients reached its primary endpoint with an ORR of 36%, showing efficacy in patients with chemoresistant disease [79]. Further trials to test this combination are ongoing (NCT04768296, NCT03896503), as well as trials testing berzosertib in combination with lurbinectedin (NCT04802174). CHK1 inhibitors have also demonstrated efficacy against preclinical models of SCLC, especially in combination with cisplatin and etoposide [120,121,169]. In cells treated with increasing doses of cisplatin to generate chemoresistance, CHK1 inhibition was able to reverse resistance and synergize with cisplatin [120,121]. However, a phase II trial of the CHK1 inhibitor prexasertib in extensive-stage patients did not demonstrate efficacy in either platinum-sensitive (PFS = 1.41 months, OS = 5.42 months, ORR = 5.2%) or refractory patients (PFS = 1.36 months, OS = 3.15 months, ORR = 0%) [80]. Indeed, some SCLC cell lines are resistant to CHK1 inhibition, and in resistant cells generated through treatment with increasing doses of prexasertib, WEE1 G2 checkpoint kinase (WEE1) upregulation was identified as a mediator of resistance [173]. WEE1 is another regulator of cell cycle checkpoints, which supports a possible combination of CHK1 and WEE1 inhibitors. WEE1 itself is an interesting therapeutic target in SCLC. A recent study identified WEE1 as a mediator of DNA repair following radiation-induced damage, with high levels of WEE1 in SCLC conferring radioresistance [174]. However, SCLC cell lines have a range of sensitivities to the WEE1 inhibitor adavosertib [175], and clinical trials of adavosertib as a monotherapy show minimal efficacy in SCLC [73,81]. A phase II trial testing adavosertib in combination with carboplatin, among other therapies, has recently concluded, awaiting results [82]. In cell lines resistant to adavosertib, upregulation of CHK1 through AXL receptor tyrosine kinase (AXL) and mTOR was observed [175], further supporting the combination of CHK1 and WEE1 inhibition. Similar to ATR/CHK1, ATM serine/threonine kinase (ATM) is critical for promoting cell cycle arrest and DNA repair in response to DNA damage. *ATM* KO sensitized an SCLC cell line to radiation, and inhibition of ATM with AZD1390 synergized with radiation both in vitro and in vivo, including in PDX models (Table 2) [122].

Another protein implicated in SCLC treatment resistance is schlafen family member 11 (SLFN11), an RNA-DNA helicase which promotes apoptosis in response to DNA damage at the S-phase checkpoint. In PDXs treated with cycles of chemotherapy, SLFN11 was greatly downregulated, while high *SLFN11* expression correlated with chemosensitivity in SCLC patient samples [142]. Indeed, low SLFN11 was found to mediate resistance to cisplatin, topotecan, and lurbinectedin [171,176,177], and a combination of ATR inhibition and lurbinectedin overcame this resistance [171], potentially by enhancing cell cycle checkpoint dysregulation. *SLFN11* expression is under epigenetic regulation and, in particular, the histone methyltransferase enhancer of zeste 2 polycomb repressive complex 2 subunit (EZH2) is responsible for silencing *SLFN11* [142]. Correspondingly, EZH2 is overexpressed in SCLC compared to other cancers [170,178], and EZH2 inhibition is able to delay chemoresistance and synergize with chemotherapy [142]. Phase I trials of EZH2 inhibitors are currently underway (NCT05353439, NCT03460977). In contrast to these data, a different cohort of PDXs showed no correlation between SLFN11 mRNA or protein levels and chemosensitivity [149], indicating that other mechanisms of chemoresistance are at play.

Several other DNA repair proteins have also been implicated in SCLC therapeutic resistance. In a pair of cell lines derived from the same tumor, CPH-54A and CPH-54B, the more resistant cell line had higher levels of DNA-dependent protein kinase catalytic subunit (DNA-PKcs) and RAD51 recombinase (RAD51), as well as a higher rate of double-strand break repair, suggesting that both non-homologous end joining (NHEJ) and homologous recombination (HR) may be involved [179]. A phase Ib/II trial of the DNA-PK inhibitor M3814 with chemotherapy was terminated due to slow patient accrual (NCT03116971). On the other hand, two of six SCLC patients in a phase Ib trial showed an objective response to the multi-targeted tyrosine kinase and RAD51 inhibitor, amuvatinib, in combination with chemotherapy [83]. Although an open-label, multicenter phase II trial did not show significant efficacy (PFS = 68 days, OS = 119 days, ORR = 17.4%), two patients who responded had elevated KIT proto-oncogene (KIT), suggesting that this may be a biomarker for patient selection [84]. In the same vein, poly (ADP-ribose) polymerase (PARP) inhibitors sensitize SCLC cell lines and xenograft models to multiple DNA-damaging therapeutics, including platinum, etoposide, temozolomide, and radiation [180,181,182]. This effect was associated with DNA-PKcs downregulation by veliparib and PARP trapping by talazoparib. Clinical trials of PARP inhibitors both alone [85,86] and in combination with chemotherapy [87,88] have yielded mixed results (Table 1), which suggests that identification of biomarkers will improve outcomes. Several ongoing studies continue to investigate combinations of PARP inhibitor with chemoradiotherapy (NCT03532880, NCT04170946, NCT04624204, NCT04728230). The growth factor receptor, insulin like growth factor 1 receptor (IGF1R), is also upregulated in SCLC, and inhibition of IGF1R was associated with nucleotide excision repair (NER) downregulation which synergized with chemoradiotherapy [160]. However, in untreated patients, no significant improvement was observed in PFS (4.6 vs. 4.7 months), OS (10.1 vs. 9.1 months), or ORR (49% vs. 43%) in a randomized phase II trial of chemotherapy with or without the monoclonal antibody targeting IGF1R, cixutumumab [75]. Similarly, a randomized, multi-institution phase II trial in relapsed patients comparing topotecan to the IGF1R inhibitor linsitinib did not show a significant improvement to OS (5.3 vs. 3.4 months, *p* = 0.71), though a benefit was seen in PFS (3.0 vs. 1.2 months, *p* = 0.0001) [89]. Recently, the RNA helicase DEAD-box helicase 4 (DDX4) was found to mediate cisplatin resistance in SCLC, and proteomic profiling identified upregulation of DNA repair with overexpression of DDX4 [183]. Finally, deletions in the mismatch repair (MMR) genes mutS homolog 2 (*MSH2*) and *MSH6* have been observed in SCLC [140], and an SCLC cell line deficient in MMR demonstrated resistance to an alkylating agent [184]. While MMR deficiencies have been observed to confer chemoresistance across several cancer types [185], evidence in SCLC is limited and more work will need to be conducted to determine the role of MMR in therapeutic resistance in this context.

### 2.5. Apoptosis Regulation

Apoptosis pathways are important in mediating response to DNA-damaging agents by promoting cell death. Therefore, tumor cells can promote therapeutic resistance and survival by upregulating anti-apoptotic pathways (Figure 2), such as through BCL2 family proteins. Indeed, SCLC has higher levels of BCL2 than NSCLC, and high BCL2 in SCLC correlates with cisplatin resistance and poor prognosis [186,187]. Targeting BCL2 family proteins like BCL2, BCL2 like 1 (BCL-xL), and MCL1 apoptosis regulator (MCL1) was found to promote apoptosis and synergize with chemotherapy and radiotherapy [187,188,189]. In addition, high BCL2 confers resistance to AURK inhibitors, and a combination of AURKB and BCL2 inhibition was effective in SCLC cell lines and PDXs [190]. However, challenges remain with BCL2-targeted therapies. Cells with low levels of BCL2 associated X (BAX) are resistant to BCL2 inhibition [188], and some studies have conversely found that low levels of BCL2 correlate with poorer outcomes [191,192]. In the latter case, since *BCL2* expression is observed to be driven by ASCL1 and higher in SCLC-A [19,28,188], it is possible that cells with low BCL2 have alternate resistance mechanisms, especially those driven by subtype transformation. Correspondingly, SCLC-A cell lines have been observed to be most sensitive to BCL2 inhibition [28], suggesting that patient stratification is important. Several phase II trials have been conducted with various BCL2 family inhibitors both alone [90,91,92] and in combination with topotecan [93,94] or standard chemotherapy [95,96], showing little activity in unselected patients, though one phase I trial saw objective response in six of seven untreated extensive-stage patients [97]. Nevertheless, new BCL2 targeted therapies for SCLC are still being investigated, and a recent phase I trial of a dual BCL2/BCL-xL inhibitor in combination with paclitaxel in 28 relapsed/refractory SCLC patients had an ORR of 25%, indicating that this therapy shows some promise, especially with patient selection (Table 1) [98].

As noted previously, *TP53* mutations are ubiquitous in SCLC. A proportion of these mutations abolish normal function, which promotes resistance to DNA-damaging therapies by attenuating apoptosis, and tumors with only missense *TP53* mutations may acquire co-alterations in related genes like *TP73* to achieve resistance [13]. Loss of p53 has been shown to confer resistance to both chemotherapy and radiotherapy, and accumulation of mutant p53 can have further gain-of-function effects on chemoresistance [193,194,195,196,197]. The p53 activator, APR-246/PRIMA-1^Met^, was shown to restore activity to mutant p53 and promote apoptosis, which demonstrated efficacy in SCLC cell lines and cell line-derived xenografts [193]. In addition, gain-of-function p53 mutations have been shown to mediate resistance to BCL2 inhibition, and heat shock protein 90 (HSP90) inhibition was able to destabilize mutant p53 and sensitize SCLC cell lines to BCL2 inhibition [195]. This combination was also found to downregulate MCL1 and nuclear factor kappa B (NF-κB) to further promote apoptosis [198]. Clinical trials of HSP90 inhibitors in combination with doxorubicin (terminated) [99] or carboplatin and paclitaxel (ORR = 0%) [100] have not demonstrated efficacy in SCLC, though other combinations, such as with BCL2 inhibition, may be more effective.

Another regulator of apoptosis is the JUN N-terminal kinase (JNK) pathway, which has been observed to have both pro- and anti-apoptotic effects in response to cell stress [199]. One group observed that JNK signaling in SCLC is generally anti-apoptotic and mediates resistance to cisplatin and UV radiation [200,201]. This suggests a potential synergy with JNK pathway inhibitors, though further work will be required to determine the efficacy of this combination.

### 2.6. Metabolism

Several studies have investigated the metabolic vulnerabilities of SCLC, as well as potential links to therapeutic resistance (Figure 1). Chalishazar et al. found that MYC-driven tumors, including chemoresistant ones, were dependent on arginine [151]. As noted previously, NOTCH and NO can promote chemoresistance through sGC [139], so considering that MYC cooperates with NOTCH in neuroendocrine-low transformation and arginine is a precursor in NO synthesis, arginine and MYC may cooperate to drive chemoresistance through sGC. SCLC cells have been found to harbor loss of argininosuccinate synthase 1 (ASS1), a critical enzyme in arginine synthesis, making them reliant on extrinsic sources of arginine. One study saw ASS1 loss in approximately half of SCLC patient samples and cell lines [202], and another observed that 83% of SCLC patient samples had low levels of ASS1 [203]. These arginine-dependent cells respond well to arginine depletion in preclinical models, by either arginine deiminase-based or arginase-based therapeutics [151,202,203,204]. A non-randomized, open-label phase II trial of arginine deiminase in ASS1-low SCLC patients was terminated due to slow patient accrual and lack of tumor response after 22 patients had been enrolled, though patient stratification based on factors such as *MYC* expression may be beneficial [101]. A phase I/II trial with arginase has also been completed, awaiting results (NCT03371979).

Another potential vulnerability of SCLC is glutamine, which functions at the nexus between the citric acid cycle and nucleotide synthesis. In SCLC, the purine synthesis pathway through phosphoribosyl pyrophosphate amidotransferase (PPAT) is favored over the citric acid cycle through glutaminase (GLS), potentially due to MYC since *PPAT* is one of its target genes [205]. Correspondingly, *PPAT* depletion reduced growth of SCLC cell lines, with similar effects from GLS overexpression. In NSCLC and ovarian cancer, some cisplatin-resistant cells have large nucleotide pools and are vulnerable to glutamine starvation and antimetabolites which interfere with nucleotide synthesis [206], though this has not been replicated in SCLC. Nevertheless, a recent study observed that SCLC cell lines treated with cycles of cisplatin to establish resistance had larger nucleotide pools and demonstrated that a combination of antimetabolites and an inhibitor of glutamine synthesis was effective in suppressing the growth of these cells both in vitro and in vivo (Table 2) [123].

Several other metabolic pathways have also been implicated in SCLC therapeutic resistance. Two studies have reported dysregulation of glucose metabolism in radioresistant SCLC. One group observed increased glucagon release and enolase 2 (ENO2) levels, suggesting increased glycolysis [207], and the other similarly observed higher glucose transporter 1 (GLUT1) levels and increased glycolytic activity [208]. Subsequent studies have shown that targeting glycolysis through inhibition of the lactate transporter, monocarboxylate transporter 1 (MCT1), is effective in preclinical models of SCLC and shows synergy with radiotherapy (Table 2) [124,209]. In addition, SCLC stem cells have higher levels of the glycolysis enzyme, 6-phosphofructo-2-kinase/fructose-2,6-biphosphatase 3 (PFKFB3) [125]. Inhibition of PFKFB3 by PFK158 reduced the levels of several stemness markers, such as CD133 and SOX2; impeded glycolysis; and attenuated tumor growth in SCLC cell line-derived xenografts. PFK158 was particularly effective in SCLC models overexpressing MYC due to the upregulation of aerobic glycolysis by MYC, consistent with the Warburg effect [210]. While the mechanism which links glycolytic activity to radioresistance in SCLC is unknown, studies in other cancer types have implicated improved redox stress control and DNA damage response [211]. The HIF-1 pathway is also known to promote both glycolysis and radioresistance, potentially linking these observations. More studies will be needed to establish the connection between the Warburg effect and radioresistance in the context of SCLC. A 2022 study investigating metabolic vulnerabilities in chemoresistant SCLC cell line-derived xenografts, established through cycles of cisplatin and etoposide treatment, identified reliance on the mevalonate–geranylgeranyl pyrophosphate (MVA-GGPS) pathway [212]. Combining chemotherapy with statins to target this pathway was effective against these xenograft models; however, clinical trials in untreated patients have not demonstrated efficacy of statins in combination with first-line chemotherapy or chemoradiotherapy (Table 1) [102,103,104]. Another study that year found increased lipogenesis in SCLC stem cells through the stabilization of fatty acid synthase (FASN) by the deubiquitinase USP13 [126]. FASN inhibition reduced lipogenesis and synergized with etoposide treatment in xenografts of the multidrug-resistant SCLC cell line H69AR (Table 2).

### 2.7. Drug Efflux Pumps

A common way through which tumor cells can develop therapeutic resistance is through the upregulation of ATP-binding cassette transporters which act as drug efflux pumps. In SCLC (Figure 1), several studies have correlated the expression of MDR1 and multidrug resistance associated protein 1 (MRP1) with chemoresistance and poor prognosis in SCLC patient samples and cell lines [213,214,215,216,217,218], and some relapse patients were found to have amplifications of *ABCC1*, which encodes MRP1 [140]. Indeed, one of the first studies to identify MRP1 used the multidrug-resistant H69AR cell line, which had a 100- to 200-fold increase in *ABCC1* mRNA levels compared to H69 [213]. Upregulation of MRP1 was identified downstream of SOX2, suggesting a potential mechanism [158]. Interestingly, irradiation of the H69 cell line to establish radioresistance led to upregulation of both MRP1 and its homolog, MRP2 [192], suggesting shared transcriptional programs linking resistance to chemotherapy and radiotherapy. As noted previously, another drug efflux pump, BCRP, was also implicated in SCLC chemoresistance downstream of YAP1 [125]. Preclinical studies of MDR1 and BCRP inhibitors have demonstrated the ability to sensitize SCLC cell lines to topoisomerase inhibitors like etoposide in vitro [219]. However, a non-randomized, open-label, multicenter phase II trial in 36 relapsed SCLC patients adding the multi-target drug efflux pump inhibitor, biricodar, to doxorubicin and vincristine only showed an ORR of 19% [105]. Upregulation of drug efflux pumps appears to be uncommon in SCLC, and further work will be needed to ascertain their role in therapeutic resistance as well as the efficacy of drug efflux pump inhibitors with other regimens.

### 2.8. Epigenetics

Epigenetic programs have been implicated in therapeutic resistance across many cancers, including SCLC, and interact with other mechanisms as well. Similarly to EZH2, HDAC inhibitors have been observed to reverse chemoresistance by upregulating SLFN11, which synergizes with TOP1 inhibition [176,177]. Furthermore, MYC-high tumors are sensitive to dual PI3K/HDAC inhibitors [148]. In addition to these roles, HDAC inhibitors have been found to synergize with cisplatin through the induction of S-phase arrest [220], and dual PI3K/HDAC inhibitors have been identified as radiosensitizers, synergizing with irradiation by maintaining open chromatin and inhibiting the repair of DNA double-strand breaks [127]. However, HDAC inhibitors are generally inactive against SCLC as a monotherapy, with ORRs of 0% [106,107,108], while combinations with chemotherapy have high toxicity [109,110], except for belinostat with chemotherapy, which demonstrated acceptable toxicity and an ORR of 57% (Table 1) [111]. The latter study noted that patients with high copy numbers of specific UDP glucuronosyltransferase family 1 member A1 (*UGT1A1*) mutations (*28 and *60) had slower clearance of belinostat, suggesting that selection against patients with these mutations will reduce toxicity. DNA methyltransferase 1 (DNMT1) inhibitors are also important in the context of SCLC, as the combined inhibition of DNMT1 and pyrimidine synthesis was found to sidestep the need for p53-mediated apoptosis by promoting terminal neuroendocrine maturation, demonstrating efficacy against chemoresistant SCLC cell lines [194]. A phase I trial combining DNMT1 and HDAC inhibitors in pulmonary and pleural malignancies, including SCLC, has been completed, awaiting results (NCT00037817).

In a screen of cell lines representing various cancer types with a lysine demethylase 1A (LSD1) inhibitor, SCLC was identified as the most sensitive cancer type, with DNA hypomethylation signature acting as a biomarker for sensitivity [221]. Genomic analyses showed that cell lines with high MYCN were more sensitive to LSD1 inhibition, while cell lines with high MYC were resistant. Subsequent mechanistic studies showed that LSD1 inhibitors act by inhibiting ASCL1, through silencing of INSM1 or derepression of NOTCH1, as well as efficacy in chemoresistant PDXs of SCLC [222,223]. These findings suggest that LSD1 inhibition would be most effective against neuroendocrine-high subtypes of SCLC, especially SCLC-A, consistent with the observation that more mesenchymal-like SCLC cells are resistant to LSD1 inhibition [224]. While an open-label, multicenter phase I trial of the LSD1 inhibitor, GSK2879552, was terminated after recruiting 29 patients due to toxicity and lack of response [112], clinical trials investigating combination therapies with LSD1 inhibitors are ongoing (NCT03850067, NCT05191797, NCT05420636), and selection for neuroendocrine-high SCLC patients may improve response (Table 1). At the RNA level, adenosine methylation to form m^6^A has been correlated with chemoresistance, with a panel of seven m^6^A regulators being predictive of survival in a cohort of 200 patients [225]. In particular, methyltransferase 3 N6-adenosine-methyltransferase complex catalytic subunit (METTL3), which is responsible for m^6^A deposition on mRNA, was found to promote chemoresistance by activating mitophagy, and inhibition of METTL3 synergized with chemotherapy in H69AR xenografts (Table 2) [128].

Aside from these, recurrent mutations in epigenetic regulation genes such as *KMT2A*, *CREBBP*, and *EP300* have been observed in SCLC [5,6,7,8,11]. A 2024 study looking at SCLC tumor evolution throughout treatment found that patients with mutations in the lysine acetyltransferase genes *CREBBP* and *EP300* have an elevated risk of disease recurrence, implicating these genes in therapeutic resistance [13]. In particular, patients with *CREBBP* mutations have an increased sensitivity to HDAC inhibition [226], suggesting this alteration as a biomarker for HDAC inhibitors. A 2020 study of the SCLC epigenome further identified several epigenetic markers which are associated with sensitivity to agents such as ATR, AURK, BCL2, and mTOR inhibitors, though these associations require further validation [227]. Interestingly, the 5′ untranslated region of *EZH2* was associated with sensitivity to AURK inhibitors and a fibroblast growth factor receptor (FGFR) inhibitor, suggesting that these therapies may be effective against SCLC cases where EZH2-mediated silencing of SLFN11 mediates chemoresistance. While mutations in other epigenetic regulation genes have been observed in SCLC, their impact on therapeutic resistance and the possibility of targeting them for treatment have not been investigated.

### 2.9. Other Resistance Mechanisms and Therapeutic Opportunities

Aside from these, several other factors have been implicated in SCLC therapeutic resistance. In both SCLC cell line-derived xenografts and PDXs, upregulation of cancer testis antigens has been observed following chemotherapy [142,228]. Cancer testis antigens are proteins normally found in male germ cells that can also have roles in cancers. In SCLC, the upregulation of two of these—PAGE family member 5 (PAGE5) and G antigen 2A (GAGE2A)—was found to promote chemoresistance, while *PAGE5* and *GAGE2A* KD cell lines were sensitized to chemotherapy [228]. Autophagy has also been linked to chemoresistance in SCLC. One group observed that upregulating autophagy, through either BMX non-receptor tyrosine kinase (BMX) signaling or sequestosome 1 (SQSTM1), mediates chemoresistance, while inhibition of autophagy using chloroquine reverses resistance and synergizes with chemotherapy [229,230]. Furthermore, JNK signaling can promote autophagy to avoid apoptosis [199], which could explain its anti-apoptotic role in SCLC [200,201]. Unfortunately, clinical trials of chloroquine and its derivative, hydroxychloroquine, have been terminated due to slow patient accrual, with one showing toxicity and lack of response (NCT00969306, NCT01575782, NCT02722369). Another study identified a mutation in the translation factor, eukaryotic translation initiation factor 3 subunit A (eIF3a), as a mediator of chemoresistance by conducting genomic analyses on pre- and post-chemotherapy samples in a cohort of 52 patients [231]. This mutation conferred chemoresistance while attenuating the growth of SCLC cell lines, similar to the effects of neuroendocrine-low transformation.

Studies have also looked at chemo- and radiosensitizers which do not target or reverse a particular resistance mechanism. One approach has been to arrest the cell cycle to sensitize cells to DNA damage (Figure 2). Patel et al. found that the microtubule inhibitor albendazole arrested SCLC cell lines in the radiosensitive G_2_/M phase, synergizing with radiation [131]. A recent study identified the cell cycle regulator cell division cycle 7 (CDC7) as a target which synergizes with chemotherapy and found that CDC7 inhibition induced cell cycle arrest at the G_1_/S checkpoint and enhanced chemotherapy-induced DNA damage [132]. Two other microtubule inhibitors, paclitaxel and vinorelbine, have also been found to synergize with cisplatin, etoposide, and radiation in vitro, reversing resistance and sensitizing cells to treatment [129,130]. However, no difference has been seen in cell cycle distribution, suggesting a mechanism other than cell cycle arrest. Unfortunately, a randomized, prospective phase III trial in 587 untreated extensive-stage patients with chemotherapy with or without paclitaxel saw an increase in toxicity with no significant improvement in failure-free survival (6.4 vs. 5.9 months, *p* = 0.179) or OS (10.6 vs. 9.9 months, *p* = 0.169) [113], though many trials continue to investigate paclitaxel in combination with various therapies (NCT02769832, NCT05420636, NCT05856695, NCT06016270). A 2022 study tested a new approach to SCLC therapy with TAK-243, an inhibitor of ubiquitin like modifier activating enzyme 1 (UBA1), an E1 enzyme [133]. TAK-243 and other inhibitors of the ubiquitin–proteasome system attenuate protein degradation and therefore have diverse effects on cells, including disruption of DNA repair and the cell cycle [232]. While the mechanism of TAK-243 in SCLC remains unclear, it has been found to synergize with both chemotherapy and radiotherapy in PDXs (Table 2) [133].

## 3. Conclusions

SCLC is a challenging malignancy to treat owing to its rapid growth, propensity to metastasize, inter- and intratumoral heterogeneity, and lack of targetable drivers. Much work has been undertaken to tackle the considerable challenge of identifying effective treatments which can overcome the widespread resistance to current therapies. Progress has been made in elucidating the biology and resistance mechanisms of SCLC, especially as it relates to the DNA damage response and SCLC differentiation and transformation. These advancements have led to new treatments with diverse approaches, many of which are currently under clinical investigation and may lead to new avenues of treatment (Table 1).

Despite this, new therapies have yet to reach the clinic in a substantial way. Given the rapid proliferation of SCLC and initial sensitivity to DNA damage, agents which target DNA damage response, such as ATR and EZH2 inhibitors, are reasonable. As well, novel therapies which target neuroendocrine-low transformation hold potential in addressing chemoresistant populations. In particular, AURKA inhibition is promising as it targets MYC and may be especially effective against EZH2-high tumors [227]. However, no single mechanism has been found to be ubiquitous in mediating therapeutic resistance, demonstrating the inter- and intratumoral heterogeneity of SCLC. This suggests that precision medicine, through biomarker detection and patient selection, is critical to delivering more effective treatment by targeting the particular mechanisms that are relevant to a particular patient. Many preclinical studies have identified biomarkers, including epigenetic markers, which influence sensitivity to certain treatments, though these can be difficult to implement and study in a clinical setting and their relevance in patients requires further investigation. Nevertheless, several clinical trials have identified potential biomarkers for previously treated patients [72,74,77,84], and the lack of response in unselected patients across several trials may be improvable through selection for these biomarkers. This is particularly relevant for resistance mechanisms which only appear in a small subset of patients, as with drug efflux pumps. Furthermore, combination therapies targeting multiple pathways may be essential to prevent the development of resistance to particular agents (Table 2).

There also remain many gaps in our understanding of resistance mechanisms and how they may be overcome. For instance, while potential biomarkers have been identified clinically, the mechanisms through which they affect the treatment response are unknown, such as in elevated KIT sensitizing to amuvatinib [84]. Similarly, though several epigenetic markers have been associated with sensitivity to certain agents, the underlying mechanisms remain to be elucidated [227]. Additionally, most efforts have focused on chemoresistance, with less emphasis placed on radioresistance, since radiotherapy is given less frequently to SCLC patients. The identification of radiosensitizers in SCLC may enable broader application of radiotherapy as a treatment modality, particularly against extensive-stage disease where evidence is mixed. In addition, factors aside from the intrinsic genetic features of the tumor, such as epigenetics and the TME, are areas which require more attention in the context of SCLC. Given how rapidly SCLC tumors are capable of recurring and developing therapeutic resistance, it is likely that faster, epigenetic mechanisms of regulation play a critical role. Indeed, recurrent mutations in epigenetic regulation genes are observed in SCLC patients, and control other key resistance factors such as MYC and SLFN11 [142,148]. However, the clinical integration of HDAC inhibitors has proven challenging due to toxicity when combined with chemotherapy and warrants further investigation to minimize toxicity. Other agents targeting DNMT1 and LSD1 also require further investigation in combination with other therapies, as SCLC is not sensitive to monotherapies targeting epigenetic regulators [106,107,108,112]. TME modulation is another area which requires further study. The immune microenvironment of SCLC has received significant attention which has led to the approval of immunotherapies against PD-1 and DLL3, highlighting the potential that these avenues of investigation may hold. Yet limited focus has been placed on other microenvironment factors, such as hypoxia and CAFs, particularly in the context of modulating therapeutic sensitivity. Further exploration of these factors will inform the feasibility of implementing TME targeted therapies, such as HIF-1 antagonists, in SCLC.

An additional challenge is the paucity of patient samples for research. This limits our ability to investigate and understand both the heterogeneity of SCLC as well as epigenetic and TME features, which are difficult to capture using other preclinical models. The incorporation of new techniques in the clinic, such as liquid biopsy, may be a promising approach to tackle some of these challenges in the future. Detecting tumor cells and DNA in the blood is a much less invasive way to probe for biomarkers, which will simplify patient selection in the context of both treatment and clinical trials. It also facilitates SCLC patient sample collection, paving the way for larger-scale studies comparing untreated and relapse samples to identify resistance mechanisms, as well as studies investigating epigenetic features. Indeed, efforts are ongoing to leverage liquid biopsy to study clonal tumor evolution and the influence of genetic and epigenetic factors on therapeutic resistance. Ultimately, advancements in the laboratory and clinic will continue to deepen our understanding of SCLC biology to surmount treatment resistance and develop more effective therapeutic approaches.

## Figures and Tables

**Figure 1 cancers-16-03438-f001:**
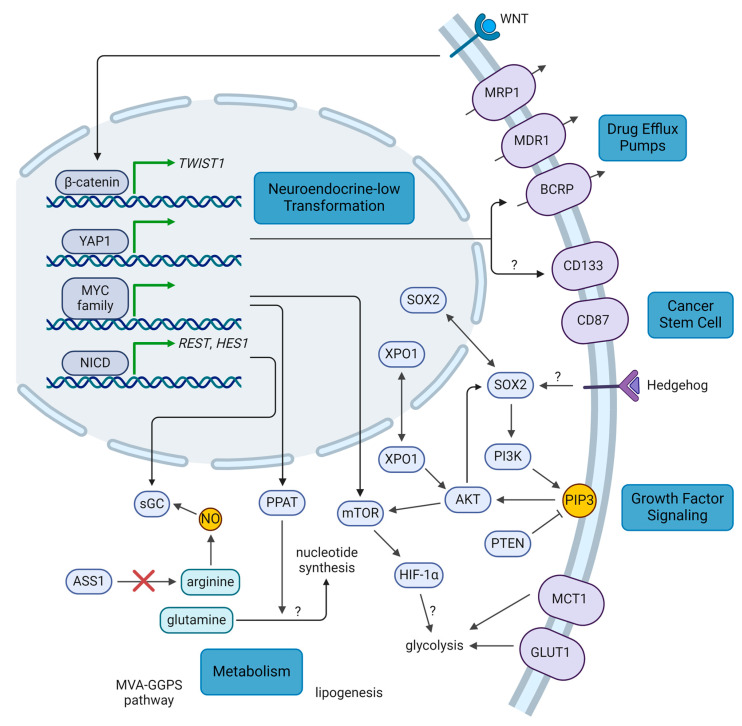
Summary of key molecular pathways in therapeutic resistance. WNT, YAP1, NOTCH, and *MYC* signaling promote transformation to a resistant, neuroendocrine-low phenotype. Cancer stem cell factors such as SOX2, CD133, and CD87 promote a resistant cell state that survives treatment. Expression of drug efflux pumps may also be associated with this cell state. Growth factor signaling pathways promote cell proliferation and therapeutic resistance. Certain resistance pathways may have metabolic dependencies that can be targeted. Question marks: mechanisms which remain uncertain or unknown. AKT: AKT serine/threonine kinase, ASS1: argininosuccinate synthase 1, BCRP: breast cancer resistance protein, GLUT1: glucose transporter 1, *HES1*: hes family bHLH transcription factor 1, HIF-1α: hypoxia inducible factor 1α, MCT1: monocarboxylate transporter 1, MDR1: multidrug resistance protein 1, MRP1: multidrug resistance associated protein 1, mTOR: mechanistic target of rapamycin kinase, MVA-GGPS: mevalonate-geranylgeranyl pyrophosphate, *MYC*: *MYC* proto-oncogene, NICD: NOTCH intracellular domain, NO: nitrous oxide, NOTCH: Notch receptor, PI3K: phosphatidylinositol-4,5-bisphosphate 3-kinase, PIP3: phosphatidylinositol (3,4,5)-trisphosphate, PPAT: phosphoribosyl pyrophosphate amidotransferase, PTEN: phosphatase and tensin homolog, *REST*: RE1 silencing transcription factor, sGC: soluble guanylyl cyclase, SOX2: SRY-box transcription factor 2, *TWIST1*: twist family bHLH transcription factor 1, WNT: Wnt family, XPO1: exportin 1, YAP1: Yes1 associated transcriptional regulator (created with BioRender.com, accessed on 4 October 2024).

**Figure 2 cancers-16-03438-f002:**
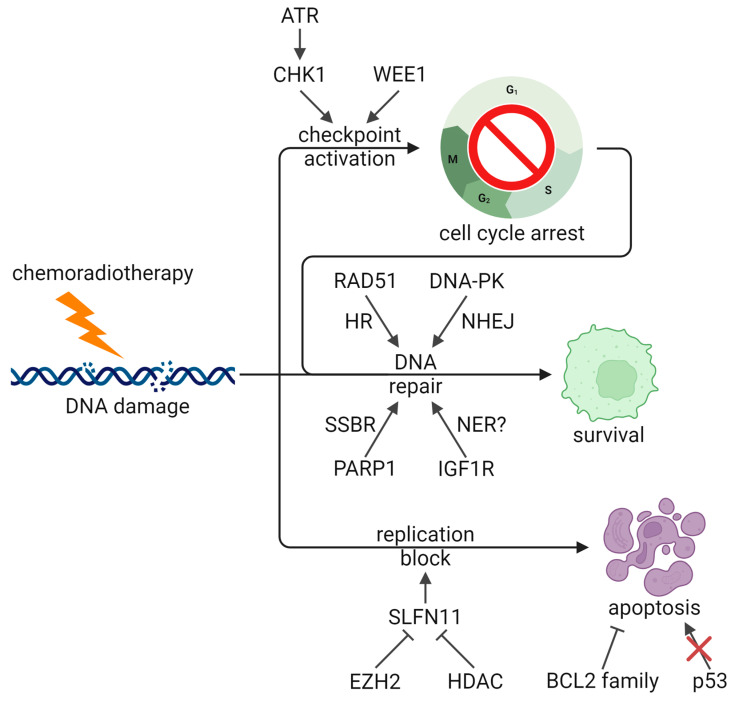
DNA damage response and apoptosis pathways in therapeutic resistance. ATR/CHK1 and WEE1 promote cell cycle arrest to promote repair, and several DNA repair pathways are implicated in promoting treatment resistance and survival. EZH2 and HDACs inhibit SLFN11-mediated replication block and apoptosis, and both BCL2 activity and *TP53* loss also block apoptosis. ATR: ATR serine/threonine kinase, BCL2: BCL2 apoptosis regulator, CHK1: checkpoint kinase 1, DNA-PK: DNA-dependent protein kinase, EZH2: enhancer of zeste 2 polycomb repressive complex 2 subunit, HDAC: histone deacetylase, HR: homologous recombination, IGF1R: insulin like growth factor 1 receptor, NER: nucleotide excision repair, NHEJ: non-homologous end joining, PARP1: poly (ADP-ribose) polymerase 1, RAD51: RAD51 recombinase, SLFN11: schlafen family member 11, SSBR: DNA single strand break repair, *TP53*: tumor protein p53, WEE1: WEE1 G2 checkpoint kinase (Created with BioRender.com, accessed 4 October 2024).

**Table 1 cancers-16-03438-t001:** Clinical investigations in overcoming SCLC therapeutic resistance. Outcomes of trials containing multiple cancer types only consider participants with SCLC.

Mechanism	Therapeutic Strategy	Treatment Regimen	Trial Phase	Participants	Outcome	NCT Number
Neuroendocrine-High to Neuroendocrine-Low Transformation	NOTCH pathway inhibition	Platinum and etoposide plus placebo vs. tarextumab	Ib/II	Untreated extensive-stage SCLC	PFS: 5.5 vs. 5.5 months OS: 10.3 vs. 9.3 months ORR: 70.8% vs. 68.6%	NCT01859741 [70]
	AURKA inhibition	Alisertib	I/II	Relapsed/refractory SCLC and other advanced solid tumors	PFS: 2.1 months ORR: 21%	NCT01045421 [71]
			II	Relapsed/refractory extensive-stage SCLC	Recruiting	NCT06095505
		Paclitaxel plus placebo vs. alisertib	II	Relapsed/refractory SCLC	PFS: 2.17 vs. 3.32 months OS: 5.42 vs. 6.11 months ORR: 18% vs. 22%	NCT02038647 [72]
	AURKB inhibition	Barasertib-HQPA	II	Relapsed/refractory SCLC	PFS: 1.6 months OS: 5.3 months ORR: 0%	NCT03366675 [73]
Cancer Stem Cells	Hedgehog pathway inhibition	Cisplatin and etoposide plus sonidegib	I	Untreated extensive-stage SCLC	PFS: 5.5 months OS: 19.7 months ORR: 79%	NCT01579929 [74]
		Cisplatin and etoposide vs. cisplatin and etoposide plus vismodegib vs. other treatments	II	Untreated extensive-stage SCLC	PFS: 4.7 vs. 4.4 months OS: 9.1 vs. 9.8 months ORR: 43% vs. 52%	NCT00887159 [75]
Growth Factor Signaling	PI3K inhibition	Cisplatin and etoposide plus buparlisib	I	Extensive-stage SCLC and other advanced solid tumors	Completed, awaiting results	NCT02194049
	mTOR inhibition	Temsirolimus	II	Extensive-stage SCLC, responding or stable disease after chemotherapy	PFS: 2.2 months OS: 8 months ORR: 1.2%	NCT00028028 [76]
		Everolimus	II	SCLC progressing after chemotherapy	PFS: 1.3 months OS: 6.7 months ORR: 3%	NCT00374140 [77]
		Paclitaxel plus everolimus	Ib	Relapsed/refractory SCLC	ORR: 28%	NCT01079481 [78]
		Vistusertib	II	Relapsed/refractory SCLC with *RICTOR* amplification	PFS: 1.2 months OS: 11.0 months ORR: 0%	NCT03106155 [73]
	XPO1 inhibition	Selinexor	II	Relapsed SCLC	Terminated—slow patient accrual	NCT02351505
DNA Damage Response	ATR inhibition	Topotecan plus berzosertib	I/II	Relapsed SCLC and other small cell cancers	PFS: 4.8 months OS: 8.5 months ORR: 36.0% Expanded trial in progress	NCT02487095 [79]
			II	Relapsed platinum-resistant SCLC	Completed, awaiting results	NCT04768296
		Topotecan vs. topotecan plus berzosertib	II	SCLC and other small cell cancers	In progress	NCT03896503
		Lurbinectedin plus berzosertib	I/II	SCLC and other small cell and high-grade neuroendocrine cancers	Recruiting	NCT04802174
	CHK1 inhibition	Prexasertib	II	Extensive-stage SCLC	Platinum-sensitive: PFS: 1.41 months OS: 5.42 months ORR 5.2% Platinum-refractory: PFS: 1.36 months OS: 3.15 months ORR: 0%	NCT02735980 [80]
	WEE1 inhibition	Adavosertib	Ib	SCLC and other advanced solid tumors	*CCNE1* and/or *MYC* family amplifications: PFS: 1.2 months ORR: 0% Others: PFS: 1.3 months ORR: 8.3%	NCT02482311 [81]
			II	Relapsed/refractory SCLC	Unselected: PFS: 1.3 months OS: 7.8 months ORR: 0% *MYC* family amplification or *CDKN2A* and *TP53* mutations: PFS: 1.2 months OS: 7.7 months ORR: 0%	NCT02593019 [73]
		Adavosertib plus carboplatin or other therapies	II	Refractory extensive-stage SCLC	Completed, awaiting results	NCT02937818 [82]
	EZH2 inhibition	Topotecan and pembrolizumab plus tazemetostat	I	Relapsed SCLC	Recruiting	NCT05353439
		PF-06821497 or standard of care plus PF-06821497	I	Relapsed/refractory SCLC and other cancers	Recruiting	NCT03460977
	DNA-PK inhibition	Cisplatin and etoposide plus M3814	Ib/II	Untreated extensive-stage SCLC	Terminated—slow patient accrual	NCT03116971
	RAD51 inhibition	Standard of care plus amuvatinib	Ib	SCLC and other solid tumors	ORR: 33%	NCT00881166 [83]
		Platinum and etoposide plus amuvatinib	II	Refractory SCLC	PFS: 68 days OS: 119 days ORR: 17.4%	NCT01357395 [84]
	PARP inhibition	Talazoparib	I	Advanced/relapsed SCLC and other solid tumors	PFS: 11.1 weeks ORR: 8.7%	NCT01286987 [85]
		Olaparib	II	Relapsed SCLC with HR deficiency	PFS: 1.4 months OS: 8.6 months ORR: 6.7%	NCT03009682 [86]
		Cisplatin and etoposide plus placebo vs. veliparib	I/II	Extensive-stage SCLC	PFS: 5.5 vs. 6.1 months OS: 8.9 vs. 10.3 months ORR: 65.6% vs. 71.9%	NCT01642251 [87]
		Carboplatin and etoposide plus placebo and placebo maintenance vs. veliparib and placebo maintenance vs. veliparib and veliparib maintenance	I/II	Untreated extensive-stage SCLC	PFS: 5.6 vs. 5.7 vs. 5.8 months OS: 12.4 vs. 10.0 vs. 10.1 months ORR: 64% vs. 59% vs. 77%	NCT02289690 [88]
		Radiotherapy plus olaparib	I	SCLC	In progress	NCT03532880
		Thoracic radiotherapy plus talazoparib	I	Extensive-stage SCLC	Recruiting	NCT04170946
		Chemoradiotherapy and pembrolizumab followed by pembrolizumab plus placebo vs. olaparib	III	Untreated limited stage SCLC	Recruiting	NCT04624204
		Chemoradiotherapy plus durvalumab and olaparib	I/II	Extensive-stage SCLC	Recruiting	NCT04728230
	IGF1R inhibition	Cisplatin and etoposide vs. cisplatin and etoposide plus cixutumumab vs. other treatments	II	Untreated extensive-stage SCLC	PFS: 4.7 vs. 4.6 months OS: 9.1 vs. 10.1 months ORR: 43% vs. 49%	NCT00887159 [75]
		Topotecan vs. linsitinib	II	Relapsed SCLC	PFS: 3.0 vs. 1.2 months OS: 5.3 vs. 3.4 months ORR: 13% vs. 0%	NCT01533181 [89]
Apoptosis Regulation	BCL2 inhibition	Bortezomib	II	Relapsed/refractory extensive-stage SCLC	PFS: 1 month OS: 3 months ORR: 4%	NCT00068289 [90]
		Gossypol	II	Sensitive relapsed extensive-stage SCLC	TTP: 1.7 months OS: 8.5 months ORR: 0%	NCT00773955 [91]
		Navitoclax	I/IIa	SCLC and other solid tumors	PFS: 1.5 months OS: 3.2 months ORR: 2.6%	NCT00445198 [92]
		Topotecan plus gossypol	I/II	Relapsed/refractory SCLC	Platinum-sensitive: TTP: 17.4 weeks ORR 17% Platinum-refractory: TTP: 11.7 weeks ORR: 0%	NCT00397293 [93]
		Topotecan plus obatoclax	I/II	Relapsed SCLC	PFS: 2 months ORR: 0%	NCT00521144 [94]
		Carboplatin and etoposide vs. carboplatin and etoposide plus oblimersen	II	Untreated extensive-stage SCLC	FFS: 7.6 vs. 6.0 months OS: 10.6 vs. 8.6 months ORR: 60% vs. 61%	NCT00042978 [95]
		Carboplatin and etoposide vs. carboplatin and etoposide plus obatoclax	I/II	Untreated extensive-stage SCLC	PFS: 5.2 vs. 5.8 months OS: 9.8 vs. 10.5 months ORR: 53% vs. 62%	NCT00682981 [96]
		Cisplatin and etoposide plus gossypol	I	Untreated extensive-stage SCLC and other solid tumors	ORR: 85.7%	NCT00544596 [97]
		Paclitaxel plus pelcitoclax	Ib/II	Relapsed/refractory SCLC	ORR: 25%	NCT04210037 [98]
	HSP90 inhibition	Doxorubicin plus ganetespib	I/II	Refractory SCLC and other solid tumors	Terminated—no significant activity	NCT02261805 [99]
		Carboplatin and paclitaxel plus SNX-5422	I	SCLC and other lung cancers	ORR: 0%	NCT01892046 [100]
Metabolism	Arginine depletion	Pegargiminase	II	Relapsed SCLC with low ASS1	Terminated—slow patient accrual and no significant activity	NCT01266018 [101]
		Pembrolizumab plus pegzilarginase	I/II	Relapsed/refractory extensive-stage SCLC	Completed, awaiting results	NCT03371979
	Statin treatment	Cisplatin and irinotecan plus simvastatin	II	Untreated extensive-stage SCLC	PFS: 6.1 months OS: 11 months ORR: 75%	NCT00452634 [102]
		Chemoradiotherapy plus placebo vs. pravastatin	III	Untreated SCLC	PFS: 7.3 vs. 7.7 months OS: 10.6 vs. 10.7 months ORR: 69.1% vs. 69.0%	NCT00433498 [103]
		Cisplatin and irinotecan vs. cisplatin and irinotecan plus simvastatin	II	Untreated extensive-stage SCLC	PFS: 6.4 vs. 6.3 months OS: 15.2 vs. 14.4 months ORR: 84.7% vs. 88.5%	NCT01441349 [104]
Drug Efflux Pumps	Drug efflux pump inhibition	Doxorubicin and vincristine plus biricodar	II	Relapsed SCLC	ORR: 19%	NCT00003847 [105]
Epigenetics	HDAC inhibition	Romidepsin	II	Refractory SCLC and other lung cancers	ORR: 0%	NCT00020202 [106]
			II	Sensitive relapsed SCLC	PFS: 1.8 months OS: 6 months ORR: 0%	NCT00086827 [107]
		Panobinostat	II	Relapsed SCLC	TTP: 1.41 months ORR: 0%	NCT01222936 [108]
		Carboplatin and etoposide plus panobinostat	I/II	SCLC and other lung cancers	Terminated—treatment toxicity	NCT00958022 [109]
		Carboplatin, etoposide, and atezolizumab plus entinostat	I	Untreated extensive-stage SCLC	Terminated—treatment toxicity	NCT04631029 [110]
		Cisplatin and etoposide plus belinostat	I	SCLC and other advanced cancers	ORR: 57%	NCT00926640 [111]
	DNMT and HDAC inhibition	Decitabine and romidepsin vs. decitabine and romidepsin plus celecoxib	I	SCLC and other pulmonary and pleural malignancies	Completed, not reported	NCT00037817
	LSD1 inhibition	GSK2879552	I	Relapsed/refractory SCLC	Terminated—treatment toxicity and no significant activity	NCT02034123 [112]
		Platinum, etoposide, and nivolumab plus pulrodemstat	Ib	Untreated extensive-stage SCLC	In progress	NCT03850067
		Atezolizumab maintenance plus bomedemstat	I/II	Extensive-stage SCLC	Recruiting	NCT05191797
	LSD1 and microtubule inhibition	Paclitaxel plus iadademstat	II	Relapsed/refractory SCLC and extrapulmonary high-grade neuroendocrine carcinoma	Recruiting	NCT05420636
Autophagy	Autophagy inhibition	Chloroquine	I	Stage 4 SCLC	Terminated—slow patient accrual	NCT00969306
		Radiotherapy plus chloroquine	I	Stage 1-3 SCLC	Terminated—slow patient accrual	NCT01575782
		Carboplatin and etoposide vs. carboplatin, gemcitabine, and hydroxychloroquine	II	Untreated stage 4 SCLC	Terminated—slow patient accrual, treatment toxicity, and no significant activity	NCT02722369
Cell Cycle	Microtubule inhibition	Cisplatin and etoposide vs. cisplatin and etoposide plus paclitaxel	III	Untreated extensive-stage SCLC	FFS: 5.9 vs. 6.4 months OS: 9.9 vs. 10.6 months ORR: 68% vs. 75%	NCT00003299 [113]
		Gemcitabine plus paclitaxel	II	Relapsed SCLC	In progress	NCT02769832
		Carboplatin, paclitaxel, and durvalumab	II	Untreated extensive-stage SCLC	Recruiting	NCT05856695
		Nelmastobart plus paclitaxel	Ib/II	Relapsed/refractory extensive-stage SCLC	Recruiting	NCT06016270

ASS1: argininosuccinate synthase 1, ATR: ATR serine/threonine kinase, AURK: aurora kinase, BCL2: BCL2 apoptosis regulator, *CCNE1*: cyclin E1, *CDKN2A*: cyclin dependent kinase inhibitor 2A, CHK1: checkpoint kinase 1, DNA-PK: DNA-dependent protein kinase, DNMT: DNA methyltransferase, EZH2: enhancer of zeste 2 polycomb repressive complex 2 subunit, FFS: failure-free survival, HDAC: histone deacetylase, HR: homologous recombination, HSP90: heat shock protein 90, IGF1R: insulin like growth factor 1 receptor, LSD1: lysine demethylase 1A, mTOR: mechanistic target of rapamycin kinase, *MYC*: *MYC* proto-oncogene, NCT: national clinical trial, NOTCH: Notch receptor, ORR: objective response rate, OS: overall survival, PARP: poly (ADP-ribose) polymerase, PFS: progression-free survival, PI3K: phosphatidylinositol-4,5-bisphosphate 3-kinase, RAD51: RAD51 recombinase, *RICTOR*: RPTOR independent companion of MTOR complex 2, SCLC: small cell lung cancer, *TP53*: tumor protein p53, TTP: time to progression, WEE1: WEE1 G2 checkpoint kinase, XPO1: exportin 1.

**Table 2 cancers-16-03438-t002:** Combination therapies under investigation to overcome SCLC therapeutic resistance.

Mechanism	Target	Therapy	Stage of Development
Neuroendocrine-High to Neuroendocrine-Low Transformation	YAP1	YAP1 Inhibitor + Chemotherapy	Preclinical [114]
	NOTCH	NOTCH Inhibitor + Chemotherapy	Clinical [70]
		HES1 Inhibition + Chemotherapy	Preclinical [115]
	WNT, PI3K	BET Inhibitor + mTOR Inhibitor	Preclinical [116]
	*MYC*	AURKA Inhibitor + Paclitaxel	Clinical [72]
Cancer Stem Cells	*MYC*, *SOX2*	STAT3 Inhibitor + Chemotherapy	Preclinical [117]
	*SOX2*	Hedgehog Inhibitor + Chemotherapy	Clinical [74,75]
Growth Factor Signaling	PI3K	PI3K Inhibitor + Chemotherapy	Clinical (NCT02194049)
		mTOR Inhibitor + Paclitaxel	Clinical [78]
		PI3K or mTOR Inhibitor + Radiotherapy	Preclinical [118]
		XPO1 Inhibitor + Chemotherapy	Preclinical [119]
DNA Damage Response	ATR	ATR Inhibitor + Topotecan	Clinical (NCT04768296, NCT03896503) [79]
		ATR Inhibitor + Lurbinectedin	Clinical (NCT04802174)
		CHK1 Inhibitor + Chemotherapy	Preclinical [120,121]
	WEE1	WEE1 Inhibitor + Chemotherapy	Clinical [82]
	ATM	ATM Inhibitor + Radiotherapy	Preclinical [122]
	SLFN11	EZH2 Inhibitor + Topotecan + Immunotherapy	Clinical (NCT05353439)
		EZH2 Inhibitor + Chemotherapy	Clinical (NCT03460977)
	NHEJ	DNA-PK Inhibitor + Chemotherapy	Clinical (NCT03116971)
	HR	RAD51 Inhibitor + Chemotherapy	Clinical [83,84]
	PARP1	PARP Inhibitor + Chemotherapy	Clinical [87,88]
		PARP Inhibitor + Radiotherapy	Clinical (NCT03532880, NCT04170946)
		PARP Inhibitor + Chemotherapy + Radiotherapy + Immunotherapy	Clinical (NCT04624204, NCT04728230)
	NER	IGF1R Inhibitor + Chemotherapy	Clinical [75]
Apoptosis Regulation	BCL2	BCL2 Inhibitor + Chemotherapy	Clinical [93,94,95,96,97,98]
	p53	HSP90 Inhibitor + Chemotherapy	Clinical [99,100]
Metabolism	Arginine	Arginase + Immunotherapy	Clinical (NCT03371979)
	Glutamine	Glutamine Synthesis Inhibitor + Antimetabolite	Preclinical [123]
	Glycolysis	MCT1 Inhibitor + Radiotherapy	Preclinical [124]
		PFKFB3 Inhibitor + Chemotherapy	Preclinical [125]
	MVA-GGPS	Statin + Chemotherapy	Clinical [102,104]
		Statin + Chemotherapy + Radiotherapy	Clinical [103]
	Lipogenesis	FASN Inhibition + Chemotherapy	Preclinical [126]
Drug Efflux Pumps	Drug Efflux Pumps	Drug Efflux Pump Inhibitor + Chemotherapy	Clinical [105]
Epigenetics	MYC, SLFN11	HDAC Inhibitor + Chemotherapy	Clinical [109,111]
		HDAC Inhibitor + Chemotherapy + Immunotherapy	Clinical [110]
		Dual PI3K/HDAC Inhibitor + Radiotherapy	Preclinical [127]
	p53, MYC, SLFN11	DNMT Inhibitor + HDAC Inhibitor	Clinical (NCT00037817)
	MYCN, ASCL1	LSD1 Inhibition + Chemotherapy + Immunotherapy	Clinical (NCT03850067)
		LSD1 Inhibition + Immunotherapy	Clinical (NCT05191797)
	MYCN, ASCL1, Cell Cycle Arrest	LSD1 Inhibition + Paclitaxel	Clinical (NCT05420636)
	Mitophagy	METTL3 Inhibition + Chemotherapy	Preclinical [128]
Autophagy	Autophagy	Autophagy Inhibitor + Radiotherapy	Clinical (NCT01575782)
		Autophagy Inhibitor + Chemotherapy	Clinical (NCT02722369)
Cell Cycle	Cell Cycle Arrest	Microtubule Inhibitor + Radiotherapy	Preclinical [129,130,131]
		CDC7 Inhibitor + Chemotherapy	Preclinical [132]
		Paclitaxel + Chemotherapy	Clinical (NCT02769832) [113]
		Paclitaxel + Chemotherapy + Immunotherapy	Clinical (NCT05856695)
		Paclitaxel + Immunotherapy	Clinical (NCT06016270)
Protein Degradation	Protein Degradation	UBA1 Inhibitor + Chemotherapy or Radiotherapy or PARP Inhibitor	Preclinical [133]

ASCL1: achaete-scute family bHLH transcription factor 1, ATM: ATM serine/threonine kinase, ATR: ATR serine/threonine kinase, AURK: aurora kinase, BCL2: BCL2 apoptosis regulator, BET: bromodomain and extraterminal domain, CHK1: checkpoint kinase 1, DNA-PK: DNA-dependent protein kinase, DNMT: DNA methyltransferase, EZH2: enhancer of zeste 2 polycomb repressive complex 2 subunit, HDAC: histone deacetylase, HES1: hes family bHLH transcription factor 1, HR: homologous recombination, HSP90: heat shock protein 90, IGF1R: insulin like growth factor 1 receptor, LSD1: lysine demethylase 1A, MCT1: monocarboxylate transporter 1, METTL3: methyltransferase 3 N6-adenosine-methyltransferase complex catalytic subunit, mTOR: mechanistic target of rapamycin kinase, MVA-GGPS: mevalonate-geranylgeranyl pyrophosphate, *MYC*: *MYC* proto-oncogene, NER: nucleotide excision repair, NHEJ: non-homologous end joining, NOTCH: Notch receptor, PARP: poly (ADP-ribose) polymerase, PFKFB3: 6-phosphofructo-2-kinase/fructose-2,6-biphosphatase 3, PI3K: phosphatidylinositol-4,5-bisphosphate 3-kinase, RAD51: RAD51 recombinase, SLFN11: schlafen family member 11, SOX2: SRY-box transcription factor 2, STAT3: signal transducer and activator of transcription 3, UBA1: ubiquitin like modifier activating enzyme 1, WEE1: WEE1 G2 checkpoint kinase, WNT: Wnt family, XPO1: exportin 1, YAP1: Yes1 associated transcriptional regulator.

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
