# Peer review of "Strategies to Target Chemoradiotherapy Resistance in Small Cell Lung Cancer"

_cancers, 2024, doi:10.3390/cancers16203438_

Round 1

Reviewer 1 Report

Comments and Suggestions for Authors  

n this review, the authors have compiled valuable information on strategies to overcome chemoradiotherapy resistance in small cell lung cancer (SCLC), a highly aggressive disease with limited treatment options and frequent relapses. Although SCLC initially responds well to therapy, it often becomes resistant upon relapse. Research has investigated various resistance mechanisms, particularly in relation to DNA damage response and cell differentiation, though many aspects, such as the roles of epigenetics and the tumor microenvironment, remain unclear. The complexity and diversity of these mechanisms suggest that no single solution exists, highlighting the need for further research to address these challenges.

This comprehensive review is both timely and informative, providing a valuable resource for a broad range of readers, including cancer biologists, oncologists, and other specialists. One minor suggestion is to present the information on combination therapies in SCLC more distinctly. While this content is already integrated throughout the manuscript, listing these therapies separately—such as Aurora Kinase Inhibitors + Chemotherapy, Temozolomide + PARP Inhibitors, Topotecan + Other Agents, and Chemotherapy + Immunotherapy—could enhance readability and make it easier for readers to access key information.

Reviewer 2 Report

Comments and Suggestions for Authors Summary of the Review Paper: The review, "Strategies to Target Chemoradiotherapy Resistance in Small Cell Lung Cancer (SCLC)," examines why SCLC resists treatment. It talks about the biology of SCLC, including genetic changes and types. The paper highlights recent findings on resistance, like DNA damage response and cancer stem cells. It also discusses current and new treatment methods, such as targeted therapies and immunotherapy, with clinical trials evaluating these methods. Strengths: Thorough Overview: The paper reviews many resistance mechanisms in SCLC, covering various biological processes. Relevant to Clinicians: It links to ongoing clinical trials, making it useful for doctors and researchers. Clear Structure: The paper is organized logically, easy to follow from biology to therapies. Updated Information: It includes recent research, especially in immunotherapy and targeted therapies. Multidisciplinary: It combines molecular biology, pharmacology, and clinical oncology. Weaknesses: Repetitive: Some parts repeat ideas, causing redundancy and lack of cohesion. No Visuals: There's a lack of diagrams for complex ideas, making it hard to understand. Limited Future Directions: The paper could explore new treatment strategies or research areas more. Brief Epigenetic Discussion: It doesn't cover epigenetic changes enough, which have therapeutic potential. Clinical Trials Focus: The paper should critically assess trial results, not just list them. Suggestions for Improvement: Remove Redundancy: Reorganize sections to avoid repeating information, ensuring a smooth flow of ideas. Add Visuals: Use figures and charts to help explain complex pathways in SCLC. Expand Epigenetic Discussion: Discuss epigenetic factors more, highlighting them as potential treatment targets. Strengthen Conclusion: Discuss future directions, new therapies, and personalized medicine in SCLC. Critically Analyze Trials: Assess clinical trials, identifying limitations and areas for improvement. Include Clinical Views: Add comments on how findings might impact SCLC treatment plans. Address Knowledge Gaps: Point out significant gaps in current research, encouraging further study. Specific Improvement Steps: 1. Remove Redundancy: Where to Improve: Sections on tumor microenvironment and DNA damage response repeat similar ideas. How to Improve: Combine these into one section discussing the tumor microenvironment's role in resistance. 2. Add Visuals: What to Add: Use figures and tables to show key resistance mechanisms. Figures: Include diagrams for DNA damage response, neuroendocrine transformation, or stem cell plasticity.

Reviewer 3 Report

Comments and Suggestions for Authors

This manuscript focus on Strategies to Target Chemoradiotherapy Resistance in SCLC. The subject is of clinical interest and the manuscript is in general well written, but often extensive and without systematization. Furthermore, some point should be clarified

1. A figure schematizing the most common alterations in SCLS and their role in cancer progression/resistance would be helpful, to illustrate the main points of the research. Also, a graphical abstract would be welcome.

2. These alterations could also be compiled in a table, referring the signalling pathway in which they are involved.

3. What is the role of Warburg effect in anticancer drug resistance in SCLC? As is one of the main metabolic alterations in cancer, this could be more exploited.

4. When the authors refer to the influence of tumor microenvironment characteristics in the mechanisms of resistance, they don’t refer to physical characteristics like hypoxia or acidity. Are these factors involved in such resistance?

5. Within all these mechanisms of resistance, what are the most important in SCLC? How can they be targeted and what are the treatment options? This should be highlighted

6. The conclusion is not well related with what was previously said, and again some systematization is needed.

Comments on the Quality of English Language

The english is in general good

Round 2

Reviewer 2 Report

Comments and Suggestions for Authors

Thank you for your detailed response to my review comments. I appreciate the effort you have put into addressing the suggestions and making the necessary revisions.

However, while I acknowledge the positive changes you’ve made, it would have been more helpful if the revisions were indicated with clear page and line numbers. Without these, it has been quite time-consuming for me to track the specific changes in the manuscript.

That said, I do appreciate the improvements, especially the consolidation of redundant sections and the expanded discussion on epigenetics and clinical trials. The inclusion of visual elements is also a valuable enhancement.